# Investigation of the Robotized Incremental Metal-Sheet Forming Process with Ultrasonic Excitation

**DOI:** 10.3390/ma15031024

**Published:** 2022-01-28

**Authors:** Vytautas Ostasevicius, Agne Paulauskaite-Taraseviciene, Ieva Paleviciute, Vytautas Jurenas, Paulius Griskevicius, Darius Eidukynas, Laura Kizauskiene

**Affiliations:** 1Institute of Mechatronics, Kaunas University of Technology, Studentu Street 56, LT-51424 Kaunas, Lithuania; ieva.paleviciute@ktu.edu (I.P.); vytautas.jurenas@ktu.lt (V.J.); darius.eidukynas@ktu.lt (D.E.); 2Department of Applied Informatics, Kaunas University of Technology, Studentu Street 50, LT-51368 Kaunas, Lithuania; agne.paulauskaite-taraseviciene@ktu.lt; 3Department of Mechanical Engineering, Kaunas University of Technology, Studentu Street 56, LT-51424 Kaunas, Lithuania; paulius.griskevicius@ktu.lt; 4Department of Computer Sciences, Kaunas University of Technology, Studentu Street 50, LT-51368 Kaunas, Lithuania; laura.kizauskiene@ktu.lt

**Keywords:** stress field, shear component, cupping test, deformation and piercing, friction force reduction

## Abstract

During the single-point incremental forming (SPIF) process, a sheet is formed by a locally acting stress field on the surface consisting of a normal and shear component that is strongly affected by friction of the dragging forming tool. SPIF is usually performed under well-lubricated conditions in order to reduce friction. Instead of lubricating the contact surface of the sheet metal, we propose an innovative, environmentally friendly method to reduce the coefficient of friction by ultrasonic excitation of the metal sheet. By evaluating the tool-workpiece interaction process as non-linear due to large deformations in the metal sheet, the finite element method (FEM) allows for a virtual evaluation of the deformation and piercing parameters of the SPIF process in order to determine destructive loads.

## 1. Introduction

Robotized single-point incremental forming (SPIF), whereby a small-sized tool deforms a sheet of metal, is ideal for replacing expensive stamping processes, assuring flexibility and cost-effectiveness, as well as increasing the competitiveness of companies. This technology opens up new possibilities compared to traditional small-series production methods wherein added value is determined by skill and qualification. The forces resulting from friction between the tool and the workpiece during machining play an important role in product quality. The influence of petroleum and vegetable-oil-based greases on the coefficient of friction, abrasion, forming forces and surface roughness of metal sheets produced by SPIF was investigated in [1]. Lubricating oil was found to produce a surface roughness close to Ra = 1.45 μm, which exceeds the surface roughness of an undeformed sheet and degrades the quality of the final product. As lubricants are not environmentally friendly, new methods for reducing and predicting forming forces need to be explored.

Various research methods have been used to evaluate the mechanical features of the robotized SPIF process [2,3,4]. The common objective of research to date [2] has been to analyse the results of SPIF processes employing industrial robots with respect to the potential increase in technological capabilities (five-axis production) and process versatility (ability to produce additional flanges or complete work) compared to traditional three-axis machining. In [3,4] Belchior et al. present an approach to link finite element method (FEM) analysis with a model of the elastic robot structure to improve the geometric accuracy of the formed metal sheet.

The complexity of SPIF mechanics and special conditions, such as bending under stress and cyclic bending/unbending, as well as the shear deformation process, which contribute to the improvement of overall formability, require innovative research efforts and techniques. Major parameters affecting formability are tool size, step down (drawing angle) and thickness and material properties of the metal sheet [5,6,7]. Formability of the metal sheet increases with decreasing tool size, step down per revolution and thickness of the sheet, while the feed rate does not have a significant effect on formability [5]. The SPIF process has much higher forming limits than those achievable by conventional sheet-forming processes, such as deep drawing, which is affected by complicated stress state and strain path. The strain-based forming limit curve criterion is widely used in the sheet-metal forming industry to predict fracture. This criterion is only valid when the strain path is linear [6]. In contrast, during the SPIF process, the strain path is rather nonlinear, and the practice of using a strain-based forming limit criterion often leads to an erroneous assessment of formability and failure prediction. Stress-based forming limits are not sensitive to strain-path changes and are used to model the necking limit, combined with the fracture limit, based on the criterion of maximum shear stress [7].

The behaviour of the ultrasonically excited forming tool during the SPIF process was investigated in [8,9]. An FEM model was employed to evaluate the influence of ultrasonic vibration frequencies and amplitudes on the accuracy of the technological process.

In order to reduce computational time in SPIF simulations, the FEM mesh is composed of several non-overlapping parts subjected to plastic and elastic deformations [10]. The plastic deformations are localized, including two substructures: plastic-non-linear and elastic-pseudo-linear. The group of plastic structures creates a finite-element mesh that is in contact with the tool used to form the material. Studies based on Marciniak–Kuczynski (MK) theory have shown that the MK damage criterion can be used to predict SPIF limits for arbitrary loading paths. An analytical process is required to investigate the extent to which such monoform strain paths can be induced by highly non-monotonic deformation pathways [11]. The MK model can be used to predict the onset of necking when strain paths have been obtained by FEM modelling. The effect of stress rate on the forming parameters of different types of steels was examined in [12]. The results revealed that the impacts of stress rate on the forces and energies used in the forming process and their limits are insignificant and highly material-dependent.

Materials such as aluminium alloys are widely used to manufacture automotive, aerospace and other components due to their good strength, low weight and formability. Therefore, the peculiarities of SPIF processes using these alloys have been extensively studied by researchers [13,14,15,16]. Based on the simplified proportional-load assumption, the fracture-boundary model, in contrast to a conventional neck, can represent the failure mode according to the AA1050-H111 failure mechanism [13]. The SPIF process applies elastic fracture criteria to the generalized tensile stress state based on the degree of triaxiality of the stresses. The research presented in [14] explains the systematic progress of SPIF formation and damage, and the extended Gurson–Tvergaard–Needleman model is used to predict surface damage. AlMgSc alloy, well known in the field of aeronautics, was investigated in [15].

Multiple experiments have been carried out wherein material laws and corresponding material variables were selected and adjusted to correctly define the behaviour of the material. In order to define the deformation behaviour of sheet metal in real production, it is essential to accurately determine the ultimate deformations of the SPIF process compared to traditional material-formation limits and the effect of the process parameters on these deformations [16]. In addition, the force of the SPIF process is very important, especially when designing special equipment or ensuring that all safety regulations are observed when using adapted machinery. Ambrogio et al. [17] examined SPIF failures by experimentally influencing process parameters, such as tool diameter, spindle rotation speed and step down on the SPIF (spifability) of AISI 304 metal sheets and analysed the results according to the circle-grid method. 

A review of the scientific literature shows that no attempt has been made so far to reduce friction between the tool and sheet metal by removing environmentally unfriendly petroleum products, which also affect the chemical composition of the formed sheets. Another important point is that the lubricated surfaces need to be cleaned after the forming process, which also increases the development time and cost of the product.

SPIF is a process in which many independent parameters operate simultaneously, and it is an important task for engineers to anticipate these factors. A review of published research results suggests that alternative methods to reduce the coefficient of friction between the tool and the metal-sheet contact surfaces have not yet been proposed; the dynamics of the process have not been sufficiently investigated, and there is a lack of physical evidence suggesting that virtual models are appropriate.

## 2. Materials and Methods

### 2.1. Methodology for Evaluation of Forming Process

Metallic materials used in deforming processes, such as single-point incremental forming (SPIF), visibly deform on contact with the tool area. The key challenge of the SPIF process is to effectively assess the shaping forces. The forces generated during SPIF may be controlled by varying the coefficient of friction between the tool and the workpiece. Lubrication is an important factor in the SPIF process, reducing friction in the tool-workpiece contact area, but the use of grease is associated with environmental problems. Therefore, it is necessary to find other ways to reduce the shaping forces associated with the process dynamics. This requires, in particular, the development of a physically adequate mathematical model of the process, the mechanical parameters of which can be adjusted experimentally using the 3D scanning device shown in Figure 1.

A 300 × 300 × 0.5 mm aluminum alloy AW1050 sheet embedded in a 30 × 30 mm L angle profile welded steel frame was formed. The frequencies of the resonant modes were determined using the COMSOL Multiphysics FEM. In order to change the lubrication of the contact surface between the forming tool and the metal sheet, an attempt was made to excite the metal sheet by vibrations. For this purpose, equipment, the scheme of which is presented in Figure 2, was developed to excite 3D ultrasonic frequency vibrations in the metal sheet.

This equipment consists of an angled-profile metal frame (2) to which piezoceramic elements (3 and 4) are glued on different planes in order to excite bending oscillations of the higher harmonics in the frame. Since the metal sheet is rigidly attached to the frame by its outer contour, the vibrations of the frame also excite the metal sheet. This principle of ultrasonic 3D vibration excitation in the metal sheet makes it possible to reduce the friction between the metal sheet and the tool during SPIF in order to improve the lubrication conditions at the tool-workpiece contact pair and, in some cases, to completely eliminate the need for lubrication. The number of piezoceramic elements in the rectangular frame used to excite the 3D vibrations in the metal sheet can range from 2 to 8, according to the given scheme (Figure 2), by alimenting them with harmonic ultrasonic-frequency electrical signals. Experimental studies on a 0.5 mm thick aluminum plate showed that the maximum reduction in friction forces at the tool-workpiece contact pair was observed when the piezoelectric actuators were excited in the 25–35 kHz frequency range.

### 2.2. Mechanical Testing of Aluminum Alloy Sheets

A 0.5 mm thick sheet of aluminum alloy, EN AW1050A H24, was used for experimental and numerical analysis. Material properties, taken from the supplier datasheet, are presented in Table 1.

As failure strain is a parameter dependent on stress state [18] and the value obtained in the uniaxial test may differ from the failure strain in the SPIF process, simple cupping was selected for calibration of the material model. During the SPIF process, the strain path is nonlinear [19]; therefore, not only the uniaxial failure strain but also the conventional forming-limit diagram criterion can cause inaccurate failure prediction.

The stress-strain relationship needed for numerical analysis was obtained by a reverse-engineering approach using data from the physical cupping test. The physical cupping test was chosen to describe the material behaviour according to the power-law material model in order to reduce the number of material-model constants needed to calibrate. Using the range of material properties from the material datasheet (yield stresses, ultimate strength range and failure strain), the power-law parameters, such as strength coefficient (K), hardening exponent (n) and failure strain, were calibrated using the LS-Opt system. The input range of the stress-strain curves for the power-law models used in LS-Opt is presented in Figure 3, where the solid line represents the calibrated power-law material curve. The power-law equation is provided below.
(1)$$\sigma =K\varepsilon^{n}=143\varepsilon^{0.097}$$
where *ε* is uniaxial strain.

### 2.3. Cupping Test for Material Model Calibration

In order to develop a validated numerical model of the SPIF process, several sets of experimental tests were performed. Firstly, the material model was calibrated by reverse engineering using experimental results of the cupping test. Subsequently, the same material model was used for the SPIF simulation. In order to characterize the plastic behaviour and ductile fracture of the aluminum alloy sheet, the Erichsen cupping test was performed on square aluminum plates, in accordance with ISO 20482: 2013 (metallic materials; sheets and strip; Erichsen tensile test). The Erichsen standard provides information on fracture under the equi-biaxial state of stress. The results are influenced by sheet thickness and the friction between the sheet and the tool surface. For the Erichsen compression test, the specimen was attached to a 55 mm die with a heavy flange and bolts tightened to achieve a force of approximately 10 kN on the workpiece holder (Figure 4). The radius of the hemi-spherical punch was 10 mm, and the test speed was 5 mm/min.

The experiment was carried out by applying a hemispherical punch to a sheet of metal until a crack appeared.

### 2.4. Experimental Investigation of SPIF Aluminum Alloy Sheet

In order to find a more efficient method to control the forming force, an attempt was made to excite the aluminum-alloy sample with ultrasonic vibrations. The experimental setup created for this purpose is shown in Figure 5. An investigated sheet (2) was clamped to the frame (1), and it was excited with two piezoelectric transducers (Ferroperm Piezoceramics A/S, Kvistgaard, Denmark) (8). A power amplifier (7) was used to generate the vibrations in the piezoelectric transducer. The robotic arm (4) incrementally formed the sheet by moving the tool (3) with the sphere attached to it. Formation force was measured by a pressing-force sensor (5) located under the frame, and values were obtained in the display controller (6).

## 3. Results and Discussion

### 3.1. Analysis of Ultrasonic Sheet Vibrations

The metal-sheet vibrational analysis, performed with the Polytec scanning vibrometer, PSV-500-3D-HV, presented in Figure 1, shows that in this frequency band, the workpiece is dominated by planar (XY) higher harmonic vibrations, which are significantly less suppressed in the tool-sheet pair friction contact than in direction Z, perpendicular to the metal sheet (Figure 6). These results show that when two piezoelectric actuators in different planes are excited, in-plane (XY) oscillations in the 30–33 kHz band are greatly enhanced.

In order to assess the effect of vibrations on the SPIF of an aluminum-alloy sheet, of surface-roughness measurements of the sheet were taken with and without ultrasonic vibrations. An advanced surface-roughness tester, TR200 (Beijing TIME High Technology Ltd., China), was used to measure roughness. The measured surface roughness of the sheet formed without ultrasonic vibrations varied in the range of R_a_ = 0.30–0.33 μm, and with ultrasonic vibrations, surface roughness was in the range of R_a_ = 0.18–0.25 μm.

In order to determine the frictional force between the tool and the aluminum-alloy sheet (Figure 7a), calculations were carried out in accordance with the diagram in Figure 7b.

Since the roughness momentum gauge was placed with an angle of 30° degrees, both pressure and friction torque could be obtained. Friction force can be calculated as follows:(2)$$F_{T}=k_{T}F_{N}$$
(3)$$F_{TO}=F_{T}\sin\alpha$$
(4)$$M_{O}=F_{O}L$$
(5)$$F_{N}=\frac{F_{O}}{\sin\alpha }$$
(6)$$M_{O}+M_{T}=\left(F_{O}+F_{TO}\right)\cdot L$$
where *L* is the length of the tool link connecting the sphere and moment gauge (150 mm); α is an angle between the sample sheet surface and the tool link (30°); *M_O_* and *M_T_* are tool-pressing-force- and tool-friction-force-generated angular momentums, respectively; *F_O_* and *F_TO_* are tool pressure force and the friction force projection to tool link, respectively; *F_N_* and *F_T_* are the tool‘s sphere pressure and friction force, respectively; *k_T_* is the friction force coefficient between the sample sheet and the steel-tool sphere; and *v_s_* is the steel tool‘s sphere speed on the surface of the sample sheet (mm/s).

The mechanical friction between the steel tool and aluminum-alloy sheet was measured using a torque sensor, STJ100, connected with a BGI series digital force gauge (Mark-10 Corp., New York, NY, USA) and a PC (Figure 7). The torque sensor‘s sensitivity factor was 6 Nm/V, and speed rate of the tool was −1200 mm/min. The measurement results of the tool friction coefficient and friction force on dry, lubricated and ultrasonically excited surfaces in the 30–33 kHz frequency range, which reduces the friction force at the tool-sheet metal contact pair, are shown in Table 2.

Table 2 shows that the coefficient of friction between the steel tool and the aluminum-alloy sheet was close to that of the lubricated surfaces when subjected to ultrasonic vibrations. This makes it possible to solve environmental problems, and the surface of the manufactured part does not need to be cleaned, while at the same time, its roughness is reduced by half. This revealed phenomenon has been patented by the authors [20].

### 3.2. Results of Theoretical and Experimental Investigation of the Aluminium-Alloy Sheet

Strain path, stress state and prediction of the formability of the SPIF were evaluated by FE analysis. In all subsequent simulations, the explicit FE code LS-Dyna was used. The aluminum sheet was modelled using fully integrated shell elements with thickness stretch allowed. Through the thickness, linear variation of strain was evaluated by five integration points, and a shear correction factor of 0.833 was used. Contact between the punch/tool and the aluminum sheet was described using the keyword: * CONTACT_FORMING _ONE_WAY _SURFACE_TO_SURFACE. There were no damage models involved; only a simple failure model was used, based on maximum failure effective strain criteria.

To simulate the cupping test and calibrate the material model. The FE model consisted of four parts: a deformable aluminum sheet, a rigid spherical punch, a bottom holder and a top holder. The holders were compressed with a force of 10 kN. The size of the shell elements varied between 0.5 at the centre/contact zone and 1 mm on the external side of the specimen. The FEM model (Figure 8a) consisted of approximately 8500 shell elements. The speed of the punch in the numerical explicit model was 1 m/s, while in physical experiments, the punch speed was 5 mm/min. The elastoplastic properties of aluminum were described by the power-law plasticity material model, and three parameters were chosen for calibration: strength coefficient (K), strain-hardening coefficient (n) and failure strain (ε_u). The simulation results are presented in Figure 8b.

After curve fitting, the experimental force-displacement relation and the FE simulation prediction (Figure 8b) correlated well. The calibrated material model curve is presented in Figure 5. The spherical punch penetration into an aluminum-alloy sheet is illustrated in Figure 9. Comparing the fracture paths also revealed a common behaviour (Figure 6b and Figure 9). The most highly stressed point is located at the apex of the dome, where the components of radial and circumferential stress and strain are equal to one another. The fracture lines are diagonal and almost symmetrically located between the centre and the corners. Three diagonal fracture lines dominate in the physical test experiments, while four diagonal fracture lines were obtained in the simulations. One of the reasons for this could be the anisotropy of the material, which is not evaluated in the FE model.

### 3.3. Numerical Simulations of SPIF Process

The SPIF model consisted of two parts: an aluminum sheet and a forming tool (Figure 10). The aluminum sheet was modelled in the same way as for the cupping test. The size of the shell elements varied between 1 mm at the centre/contact zone and 5 mm on the external side of the specimen. The FE model consisted of about 20,000 shell elements. The main features of the FE model were copied from the calibrated cupping-test model. The elastoplastic properties of aluminum were described by a power-law plasticity-material model calibrated with a cupping test. The forming tool was modelled as a rigid body using shell elements. The aluminum-sheet holder was simulated with fixed nodes (25 mm wide around all four edges). Major diameter of the helix was 140 mm.

The load was applied by three functions of tool displacements: f(x), f(y) and f(z). All functions were continuous and varied so that the horizontal displacement per revolution was 0.5 mm and the vertical displacement per revolution was 0.5 mm. In an explicit numerical model, the speed of the forming tool was about 2 m/s; in physical experiments, it was 1.2 m/min. The trajectories of the forming-tool path are shown in Figure 11.

The experimentally obtained ultrasonic excitation effect on friction forces was evaluated numerically by applying different friction coefficients. Various simulations were carried out, with coefficients of friction ranging from 0 to 0.5. The effect of friction on the horizontal-force component is shown in Figure 12.

It can be seen that increasing the coefficient of friction from 0 to 0.5 increases the amplitudes of the X-force component by a factor of approximately two. In contrast, the effect on the vertical Z-force component is negligible (Figure 13a). The effect of friction on the resultant force is shown in Figure 13b.

SPIF is characterized by a reduction in the wall thickness of the final finished part compared to the initial thickness of the sheet metal. The excessive-thickness reduction ratio in the deformation zones when the sheet metal is formed separately has a significant effect on the forming limit. Prediction of the thickness of the deformation zone is an important approach to control the thinning ratio. With regard to the object of study, aluminum alloy, the principle of thickness deformation in the SPIF process is presented in Figure 14. The relationship between wall thickness and drawing angle (α) can be expressed by the sine law used in SPIF:(7)$$t_{f}=t_{0}\sin\left(90-\alpha \right)$$
where *t*_0_ is the initial thickness; *t_f_* is the final thickness; and α is the drawing angle between the initial flat surface and the deformed surface.

The drawing angle obtained in the simulation varied from 47°; therefore, according to the sine law, this would lead a reduced thickness by up to 0.34 mm. In contrast, the minimum thickness obtained in the simulation was 0.335 mm. Three elements (A, B, C) with different drawing angles of 47°, 30° and 17°, respectively, after a forming depth of 25 mm show the development of the thickness reduction. The results obtained from the SPIF simulation correlate well with the sine law. Strain distribution after a forming depth of 25 mm is presented in Figure 15a. Shell element 6285 of the aluminum sheet comes into contact with the forming tool at approximately 35 revolutions. Figure 15b shows that a significant increase in effective strains starts after contact with the tool and increases with further revolutions of the tool.

Figure 14 and Figure 15 show that the decrease in sheet thickness during SPIF correlates with effective plastic strain changes. The thinning of the forming sheet is one of the main failure modes of SPIF and is related to the drawing angle (α), which is one of the geometrical limits of the SPIF process. Figure 15 shows that the effective strains achieved during incremental forming (maximum effective strain, 60%) are much higher than the values given in the material datasheet since tensile failure strain constitutes 3–8%, and the values of failure strain calibrated by the cupping test constitute 38%. This confirms that failure prediction in the incremental forming process is not so straightforward and that other failure criteria should be applied. Comparing the evolution of stress triaxialities during the cupping test and SPIF (Figure 16), it can be observed that during the cupping test, the stress triaxiality is stable and equal to the 2/3, which corresponds to the equi-biaxial tension stress state, while the stress triaxiality during the SPIF process varies from −0.6 to 0.6. This confirms that the stress state during the SPIF process varies from plane-strain compression up to plane-strain tension. During the SPIF process can be achieved higher formability because at the plane-strain tension state, the failure strain of the material has a lower value compared with failure strains at the other stress states.

Deformation phenomena during incremental formation are still under discussion. In the absence of direct experimental evidence, some researchers have claimed that deformation is caused by stretching, while the others state that it is caused by through-thickness shear [21,22]. If out-of-plane shear dominates, the principal strains are not in the plane [23]; therefore, the use of shell elements can be limited, as shell elements typically do not allow for capture of through-thickness shear properties in simulations. Fully integrated shell elements with a thickness stretch in LS-Dyna allow for the capture of through-thickness shear stresses and shear strains. Three elements (S2681, S2683 and S2685; see Figure 15a) along the x-axis were selected to analyse stresses and strains during simulation of the SPIF process. The strain distribution through the thickness is shown in Figure 17.

The results show that the through-thickness shear strains have significant values compared to the in-plane (x,y) shear strains. In particuar, the transverse through-thickness deformation of element No. 6285 accounts for about 50% of the in-plane shear deformation.

As vibrational excitation of the sheet-metal sample was performed by piezo actuators attached to the steel frame, the calculations were also performed by exciting the frame in which the aluminum-alloy sheet was mounted. The results of experimental and theoretical studies for the first embedded sheet in steel-frame mode, presented in Figure 18, show a good correlation between the dynamic properties of the COMSOL Multiphysics (81.707 Hz) and experimental (78.8 Hz) models.

The force-displacement response was obtained by plotting the reaction force on the punch vs. the displacement of the punch (Figure 19).

The maximum load for all five cupping tests was *F_max_* = 1160 N, Erichsen index IE = 5.36 mm. Force vs. displacement curve was chosen as the validation criterion for the numerical model.

### 3.4. Experimental Validation of SPIF Simulation Results

The SPIF experiment was performed under dry, lubricated and vibration-excited friction on the contact surfaces of the tool and the sheet metal. The vertical force dependences on the process conditions are given in Figure 20.

Accordingly, to verify modelling results (presented in Figure 8), the puncture forces of the sheet metal during experimental investigation are shown and Figure 21. It can be observed that at a depth of 0, the vertical force is equal to 224.04 N; at a depth of 10, the vertical force is 778.04 N; at a depth of 16, the vertical force is 1064 N; however, when the depth reaches a value of 15, the vertical force starts to decrease significantly.

Based on the obtained results, curve and magnitude of puncture force correlated well with modelling results; therefore, the theoretical model is verified.

In order to validate the adequacy of the simulated reduction in sheet thickness (Figure 14) for the experimentally obtained SPIF profile, the wall thickness of the formed aluminum specimen was measured at several points of the cross-sectional cut (Figure 22) using a Topex 31C629 micrometer with an accuracy of 0.01 mm.

The wall thickness of the incrementally formed aluminum sheet at measurement points is presented in Table 3.

### 3.5. Discussion

The SPIF experiment was performed under dry, lubricated and vibration-excited friction on the contact surfaces of the tool and the sheet metal. As depicted in Figure 20, depending on the forming depth, the forming force is the lowest under ultrasonic excitation. The results in Figure 6 show that when two piezoelectric actuators excite the aluminum-alloy sheet in different planes, the in-plane (XY) oscillations of the sheet in the 30–33 kHz ultrasonic frequency band are greatly amplified. Comparing the ultrasonic excitation frequencies obtained above with the simulated COMSOL Multiphysics finite elements and the experimentally obtained eigenfrequencies of the aluminum sheet in the direction perpendicular to its plane, which are 81,707 Hz and 78.8 Hz, respectively (Figure 18), these ultrasonic frequencies are several tens of times higher. The coefficient of friction of the applied concentrated contacts between the forming tool and the surface of the sheet is almost identical to the coefficient of friction between the vibrating sheet surface and the tool (Table 2). It was found that the surface roughness of the formed sheet without ultrasonic excitation varied in the range of Ra = 0.30–0.33 μm, whereas with ultrasonic excitation, it varied in the range of Ra = 0.18–0.25 μm. These SPIF properties of ultrasonic excitation could offer a better solution in terms of environmental protection, product cost and time to market compared to lubrication, which is widely used today. Measurements of the cross section of the formed sheet showed that the most deformed sheet was the thinnest at cross-sections points no 2 and 4 (Figure 22).

## 4. Conclusions

In this paper, numerical and experimental methods were used to analyse the deformation forces of single-point robotized incremental forming of an aluminum alloy sheet. We proposed an innovative method for reducing the frictional force between the forming tool and the sheet surface by excitation of the workpiece with high-frequency oscillations in two orthogonal directions in the sheet plane. The coefficient of friction between the dry sheet surface subjected to ultrasonic vibrations and the tool was found to be close to the coefficient of friction between the lubricated surfaces, which can reduce the product’s time to market and make the process more environmentally friendly. Thanks to the numerical FEM, the decreasing tendencies of the sheet cross section, depending on the degree of plastic deformation and the processes of sphere penetration into the aluminum-alloy sheet, were investigated and validated. The effective strains achieved during the incremental forming (maximum effective deformation, 60%) were found to be much higher than the values given in the material datasheet since tensile failure strain constitute 3–8%, and the values of failure strain calibrated by the cupping test constitute 38%. This confirms that failure prediction in the incremental forming process is not straightforward.

## Figures and Tables

**Figure 1 materials-15-01024-f001:**
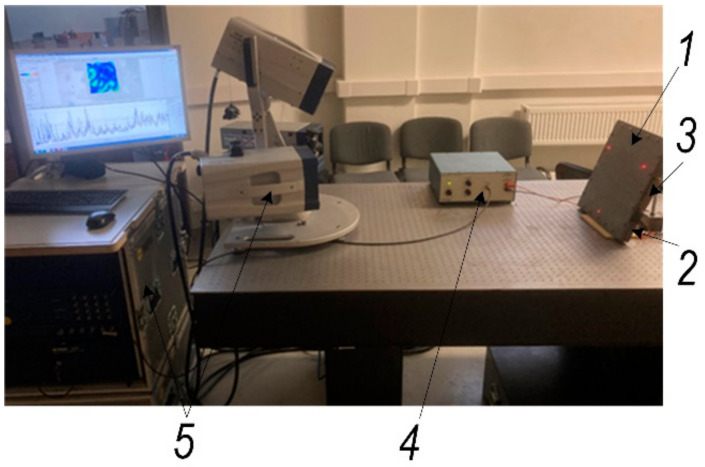
3D scanning experimental setup for the investigation. 1: an experimental body—aluminium alloy sheet; 2: steel base frame; 3: piezoelectric actuator; 4: liner amplifier P200 (FLC Electronics AB, Sweden); 5: 3D scanning vibrometer PSV-500-3D-HV (Polytec GmbH, Germany).

**Figure 2 materials-15-01024-f002:**
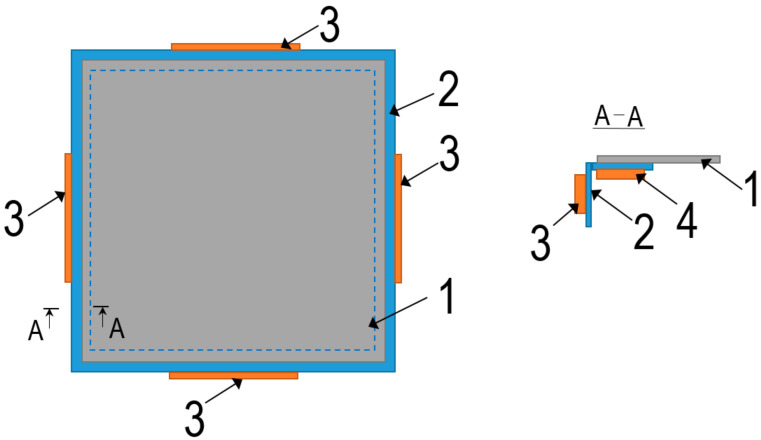
Scheme of excitation of ultrasonic 3D vibrations in the sheet. 1: metal sheet; 2: frame; 3,4: bimorph-type piezoelectric actuators.

**Figure 3 materials-15-01024-f003:**
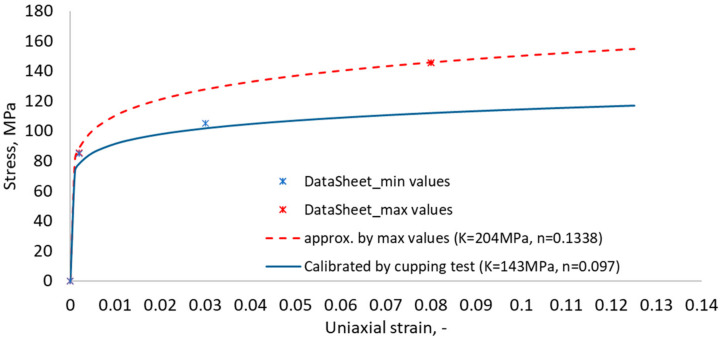
Stress-strain curves of the power-law model (dashed curves represent min and max range for calibration of the material model, and the solid line represents the calibrated power-law function).

**Figure 4 materials-15-01024-f004:**
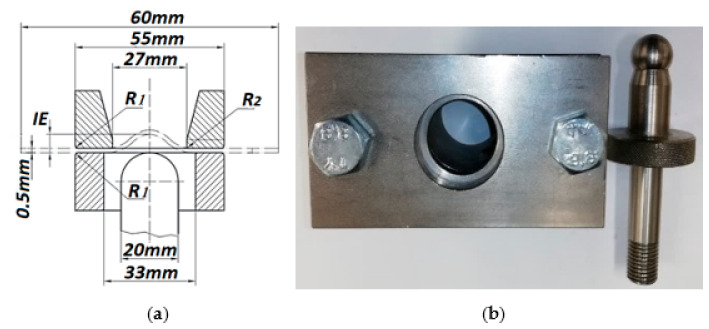
Erichsen cupping test: (**a**) scheme, (**b**) test die.

**Figure 5 materials-15-01024-f005:**
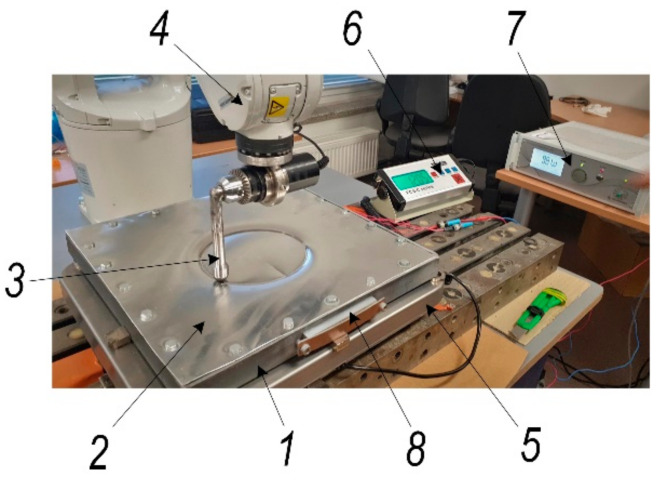
Experimental setup of incremental aluminum-alloy sheet forming. 1: frame for aluminum-sheet fixation; 2: aluminum-alloy sheet; 3: forming tool; 4: robot ABB IRB1200; 5: pressing-force sensor, FCS-4035-150; 6: controller of pressing-force sensor; 7: power amplifier; 8: piezoelectric actuators for ultrasonic excitation.

**Figure 6 materials-15-01024-f006:**
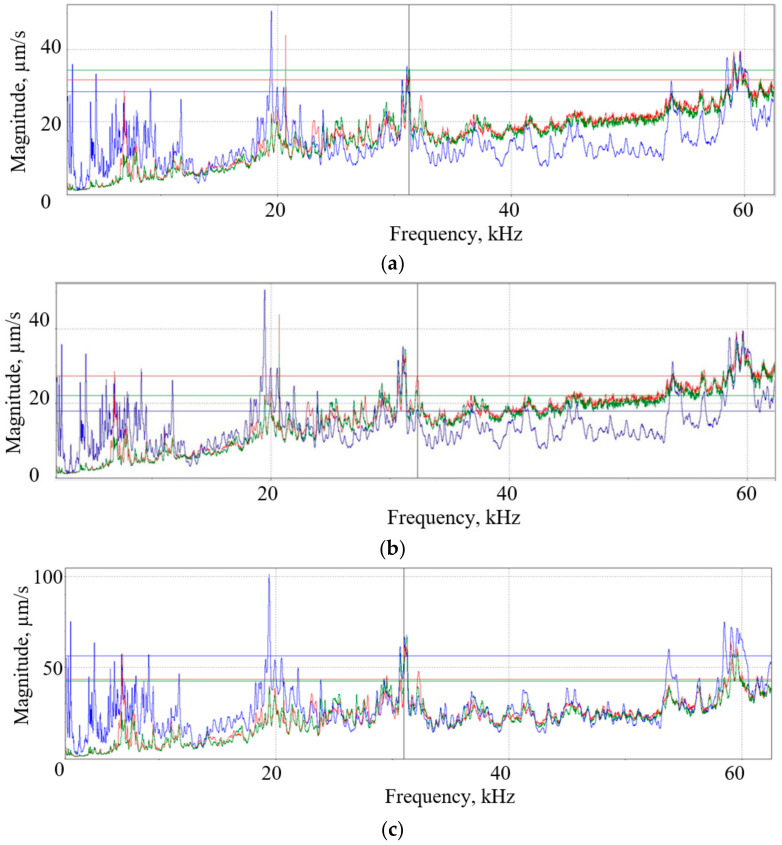
Amplitude-frequency characteristics of the metal sheet measured with a Polytec 3D scanning vibrometer in the frequency range from 2 kHz to 60 kHz after excitation of two differently arranged piezoelectric actuators (according to Figure 2): (**a**) actuator 4 is excited; (**b**) actuator 3 is excited; (**c**) both actuators are excited, where X (red) and Y (green) represent lateral vibrations and Z (blue) represents vibrations in normal direction.

**Figure 7 materials-15-01024-f007:**
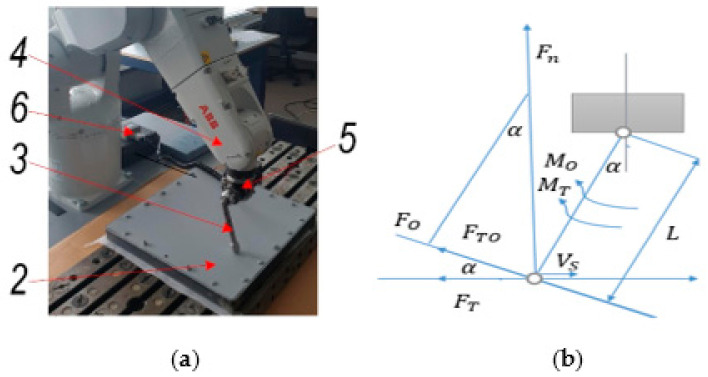
Experimental setup for investigation of friction force: (**a**) interaction and (**b**) calculation scheme of single-point tool in contact with aluminum alloy sheet. 1: frame for sheet; 2: aluminum sheet; 3: forming tool; 4; ABB IRB1200 robot; 5: mechanical torque sensor, STJ100; 6: BGI series digital force gauge.

**Figure 8 materials-15-01024-f008:**
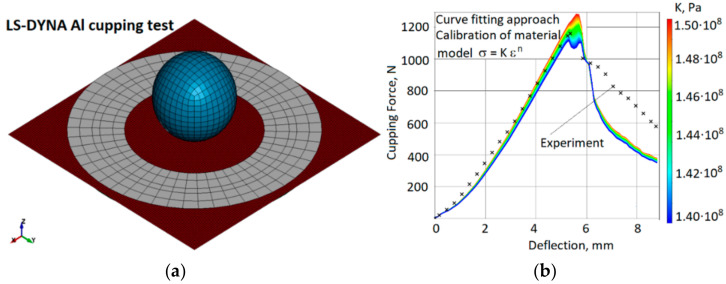
Modelling of the cupping test: (**a**) numerical model of the cupping test, (**b**) force-displacement plot of the cupping experiment and sensitivity of the simulation results to the strength coefficient (K) of the calibrated power-law material model (strain-hardening coefficient, *n* = 0.097; failure strain, *ε_u_* = 0.38).

**Figure 9 materials-15-01024-f009:**
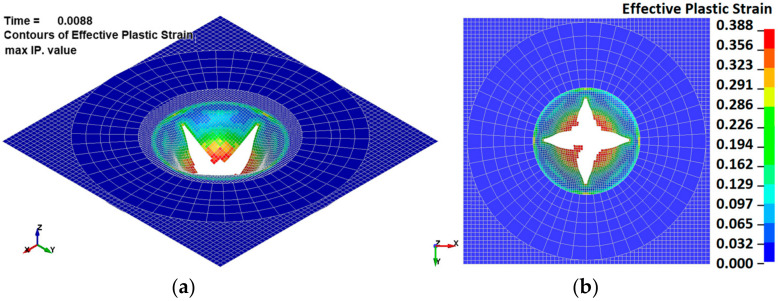
Simulated images of a spherical punch piercing an aluminum-alloy sheet during the Erichsen test: (**a**) isometric view, (**b**) at the bottom of the sheet.

**Figure 10 materials-15-01024-f010:**
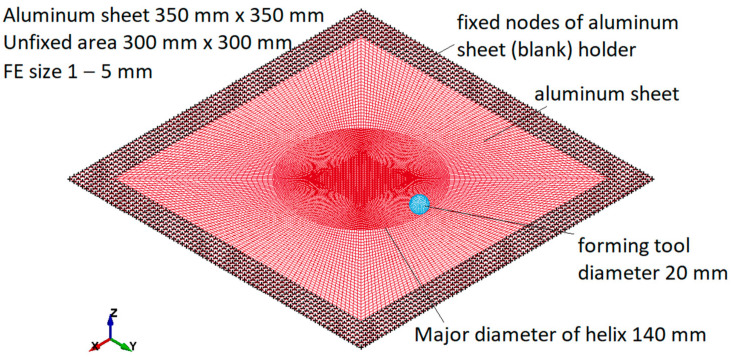
SPIF simulation model.

**Figure 11 materials-15-01024-f011:**
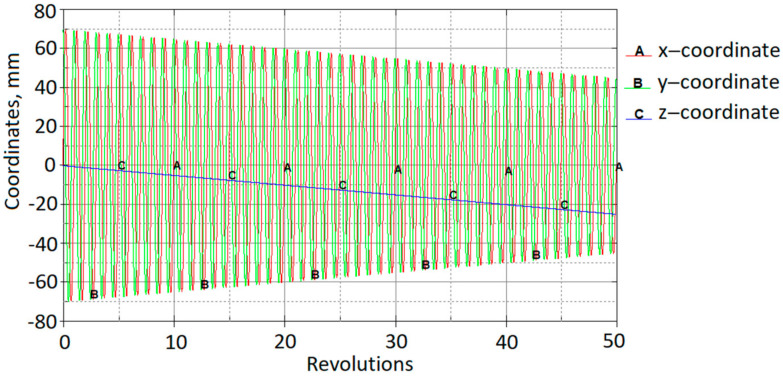
Loading-curve displacements in XYZ directions.

**Figure 12 materials-15-01024-f012:**
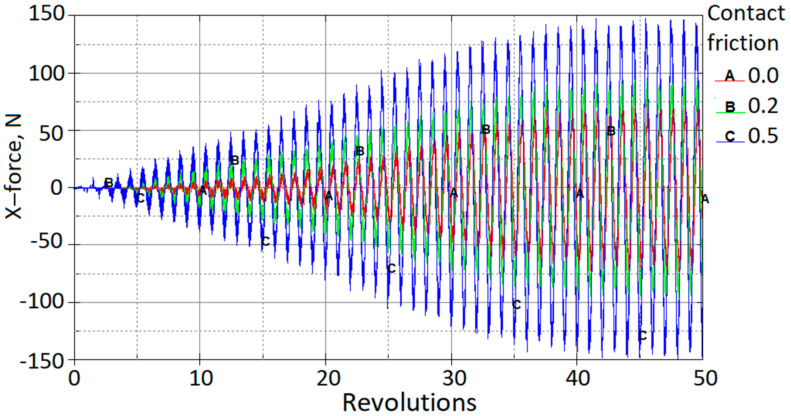
X-contact force vs. rotations at the different contact friction coefficients.

**Figure 13 materials-15-01024-f013:**
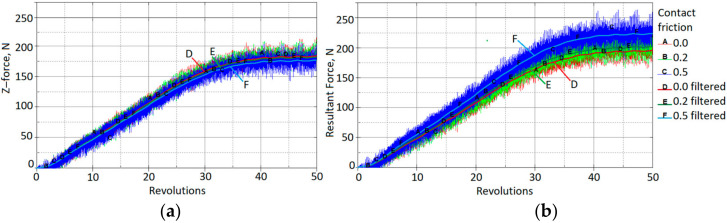
Contact force vs. rotations for different coefficients of contact friction: (**a**) z-force (vertical force), (**b**) resultant force.

**Figure 14 materials-15-01024-f014:**
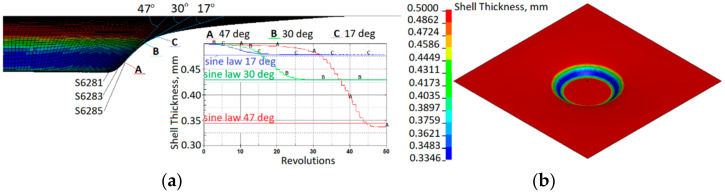
Reduction in shell thickness: (**a**) variation at elements, which, after a forming depth of 25 mm, has drawing angles of 47°, 30° and 17°; (**b**) distribution of shell thickness after a forming depth of 25 mm.

**Figure 15 materials-15-01024-f015:**
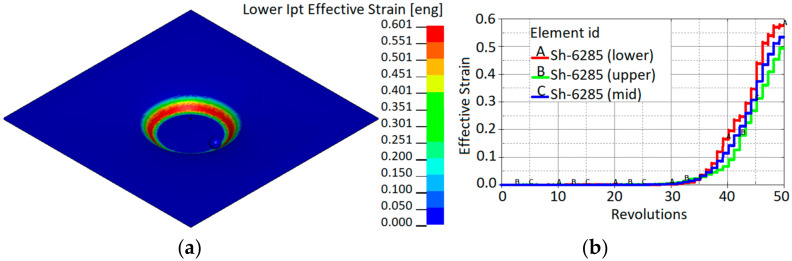
Effective strain: (**a**) distribution after a forming depth of 25 mm; (**b**) evolution in shell element 6285 during the SPIF simulation process.

**Figure 16 materials-15-01024-f016:**
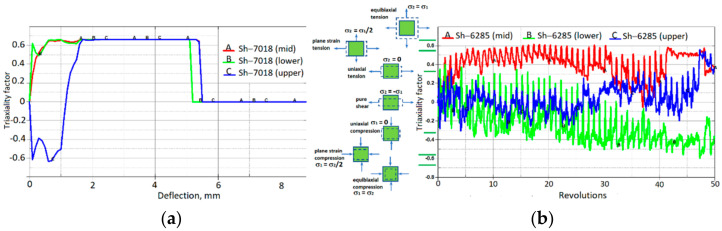
Stress-triaxiality evolutions in particular elements that were in contact with tools during the forming process: (**a**) cupping test; (**b**) SPIF process.

**Figure 17 materials-15-01024-f017:**
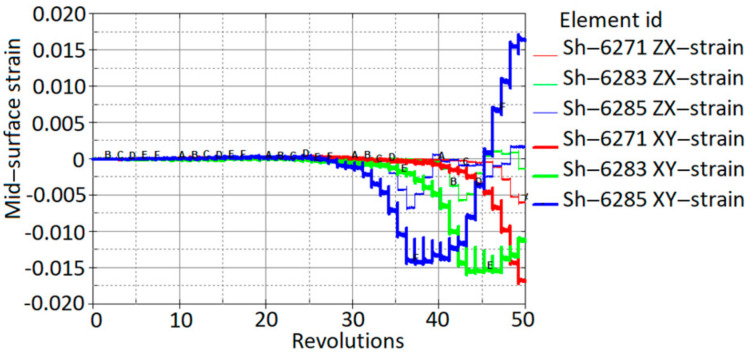
Variation of in-plane and out-of-plane shear strains during the SPIF process.

**Figure 18 materials-15-01024-f018:**
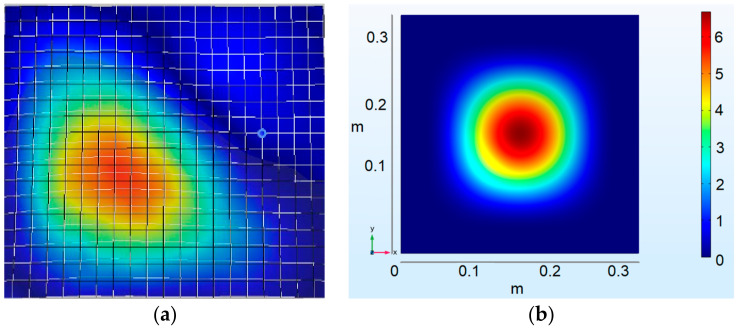
The first modes of transverse oscillations of an aluminum-alloy sheet fixed in a steel angle: (**a**) measured at a frequency of 78.8 Hz and (**b**) calculated at a frequency of 81.707 Hz.

**Figure 19 materials-15-01024-f019:**
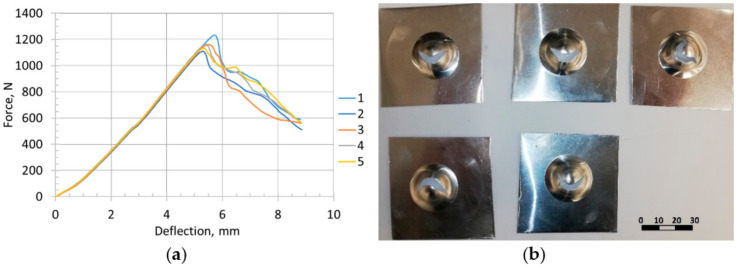
Erichsen cupping test results: (**a**) cupping curves for specimens 1–5; (**b**) nature of failure of specimens 1–5 after the cupping test.

**Figure 20 materials-15-01024-f020:**
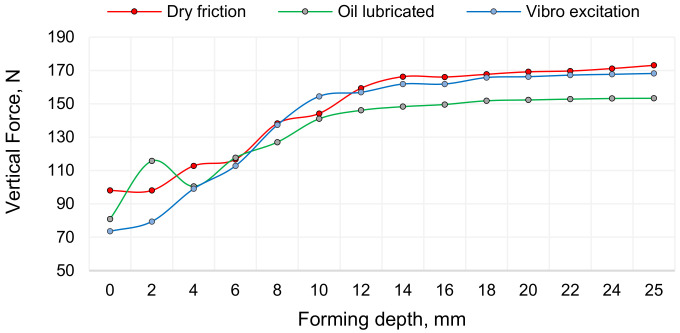
Graphical representation of vertical force dependency from different frictions when sheet thickness is 0.5 mm.

**Figure 21 materials-15-01024-f021:**
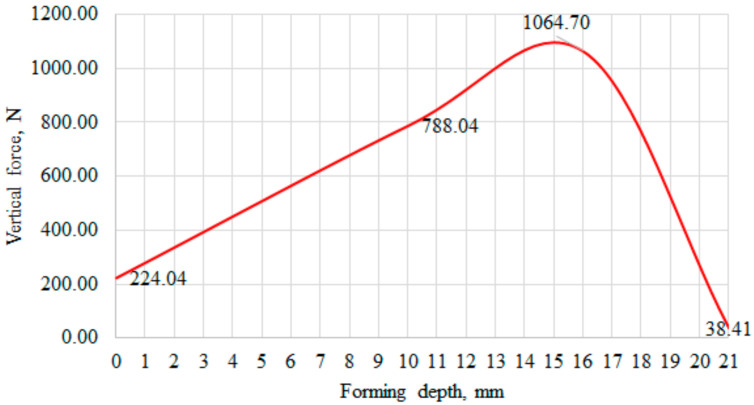
Graphical representation of the puncture force of an aluminum alloy sheet.

**Figure 22 materials-15-01024-f022:**
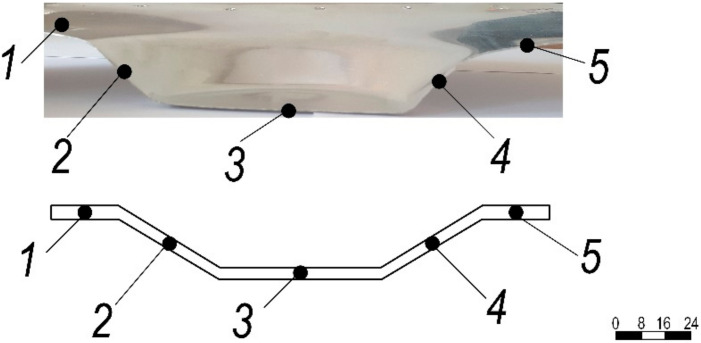
Section of incrementally formed aluminum specimen with schematic representation of measurement points.

**Table 1 materials-15-01024-t001:** Mechanical properties of aluminum alloy EN AW1050A H24 from datasheet.

	Modulus of Elasticity, GPa	Proof Stress, MPa	Tensile Strength, MPa	Elongation A, %	Strength Coefficient K, MPa	Strain-Hardening Coefficient n
Datasheet Values	71	>85	105–145	3–8		
Range for curve fitting				2 ÷ 40	140 ÷ 200	0.05 ÷ 0.2
Calibrated values LS-Opt & LS-Dyna				37.9	143	0.097

**Table 2 materials-15-01024-t002:** Friction coefficient and friction-force measurement results.

Parameter	Without Lubrication and Vibration	With Lubrication	With Vibration
Friction Force, N	3.15	1.63	1.87
Friction coefficient	0.5	0.1	0.12

**Table 3 materials-15-01024-t003:** Thickness of incrementally formed aluminum sheet at measurement points.

Measurement Point	Wall Thickness, mm	Δ from Initial Thickness, mm
1	0.51	0.01
2	0.36	0.14
3	0.49	0.01
4	0.37	0.13
5	0.51	0.01

where Δ is the deviation of the deformed sheet cross section. The results of measurement and modelling correlate with one another.

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
