# Peer review of "Investigation of the Robotized Incremental Metal-Sheet Forming Process with Ultrasonic Excitation"

_materials, 2022, doi:10.3390/ma15031024_

Round 1

Reviewer 1 Report

This manuscript is well written and designed. Only feel a discussion section comparing this processing with other metal processing methods must be provided

Author Response

Dear reviewer, thank you for your time and quality evaluation of the publication. We are sending answers to your questions.

Reviewer 2 Report

This is an interesting publication where the authors study the effect of ultrasonic excitation on the Incremental Forming Process. On the basis of experimental and theoretical results, the authors show very convincingly that the use of ultrasonic excitation makes it possible to form sheets without the traditional use of a lubricant. Also interesting is the result of obtaining high effective strains without failure. The reviewer fully agrees that the influence of the scheme and the history of deformation has a significant impact on the exhaustion of the plasticity resource in this case. The article is well written, and its results are of great scientific and practical interest. Based on this, the reviewer recommends the article for publication as it is. One small recommendation: the authors should, if possible, remove the text from the figures in the figure caption.

Author Response

(The authors gave the same response as above.)

Reviewer 3 Report

The paper "Ultrasonically Excited Metal Sheet Robotized Incremental Forming Process Investigation" presents the experimental investigation and finite element simulation of the forming process with additional ultrasonic influence. The authors have shown the positive influence of ultrasonic treatment on the friction coefficient and deformation behaviour of the aluminium sheet during cold stamping. The paper is well written and may be accepted for publication. However, some points of the paper should be clarified accordingly following comments:

  1. The authors wrote that “in case of the cupping test the 166 biaxial tension is dominant and the failure strain is larger than in the uniaxial tensile test”. It is not correct. The stress scheme in the case of the biaxial tension is harder than in the case of uniaxial tension. Stress triaxiality coefficient has a value of 0.66 and 0.33 in case of biaxial and uniaxial tension, correspondently. In the case of biaxial tension, the fracture proceeds at significantly lower values of strain. The difference between the datasheet and the calibrated values is described by the fact that the overall elongation is given in the Data Sheet without necking consideration. The significant deformation before fracture proceeds in the neck of the sample. That’s why, for finite element simulation, it is recommended to include in the calculations the energy-based and critical stress-based fracture criteria depending on the stress triaxiality coefficient (please, see 10.1016/j.ijmecsci.2016.02.001, 10.3390/app11073204). This approach may give significantly accurate results.
  2. The speed of the punch was 1-2 m/s. It is a very large speed, which may significantly increase the temperature of the aluminium sheet. However, the authors did not consider this fact. It is recommended to show the calculated temperature distribution in the aluminium sheet to approve that the temperature rise has no large influence on the deformation process. In another case, it is needed to include to the model the influence of the temperature change.
  3. Why do the imprints after the deformation using SPIF mode in Figures 14 and 19 have a non-spherical shape if the shape of the punch is spherical?
  4. The details of the finite element simulation of the ultrasonic influence should be added to the manuscript. It is unclear, how were simulated fluctuations with such high frequency.
  5. Minor corrections:
  • The position of the proof stress values in Figure 3 is not correct. Proof stress values should be shifted on the stress-strain curves to a strain of 0.002 from the finish of the elastic mode.
  • It is better to add the scale bars to Figures 19b and 22.

Author Response

(The authors gave the same response as above.)

Round 2

Reviewer 3 Report

The authors have answered previous comments and made necessary modifications to the manuscript. The paper may be accepted for publication.